# Young Artistic Gymnasts Drink Ad Libitum Only Half of Their Fluid Lost during Training, but More Fluid Intake Does Not Influence Performance

**DOI:** 10.3390/nu15122667

**Published:** 2023-06-08

**Authors:** Costas Chryssanthopoulos, Georgios Dallas, Giannis Arnaoutis, Eirini Charikleia Ragkousi, Georgia Kapodistria, Ioannis Lambropoulos, Ionas Papassotiriou, Anastassios Philippou, Maria Maridaki, Apostolos Theos

**Affiliations:** 1Department of Physiology, Medical School, National and Kapodistrian University of Athens, 11527 Athens, Greece; tfilipou@med.uoa.gr; 2School of Physical Education and Sports Science, National and Kapodistrian University of Athens, 17237 Athens, Greece; gdallas@phed.uoa.gr (G.D.); eirin1_95@hotmail.com (E.C.R.); ggeorgiakap@gmail.com (G.K.); iwn.papas@gmail.com (I.P.); mmarida@phed.uoa.gr (M.M.); 3School of Health Science and Education, Department of Nutrition and Dietetics, Harokopio University, 17671 Athens, Greece; garn@hua.gr; 4Biomedicin, Diagnostic and Research Laboratories, 15124 Marousi, Greece; j.p.lampropoulos@gmail.com; 5Section of Sports Medicine, Department of Community Medicine & Rehabilitation, Umeå University, 901 87 Umeå, Sweden; apostolos.theos@umu.se

**Keywords:** fluid balance, gymnastics, children, dehydration, exercise, urine specific gravity

## Abstract

To examine the effect of the fluid balance on and performance in young artistic gymnasts during training under ad libitum and prescribed fluid intake conditions, eleven males (12.3 ± 2.6 years, mean ± SD) performed two 3 h identical training sessions. Participants ingested, in a random order, water equivalent to either 50% (LV) or 150% (HV) of their fluid loss. After the 3 h training, the gymnasts performed program routines on three apparatuses. The pre-exercise urine specific gravity (USG) was similar between conditions (LV: 1.018 ± 0.007 vs. HV: 1.015 ± 0.007; *p* = 0.09), while the post-exercise USG was lower in the HV condition (LV: 1.017 ± 0.006 vs. HV: 1.002 ± 0.003; *p* < 0.001). Fluid loss corresponding to percentage of body mass was higher in the LV condition (1.2 ± 0.5%) compared to the HV condition (0.4 ± 0.8%) (*p* = 0.02); however, the sums of the score performances were not different (LV: 26.17 ± 2.04 vs. HV: 26.05 ± 2.00; *p* = 0.57). Ingesting fluid equivalent to about 50% of the fluid lost, which was the amount that was drunk ad libitum during training, maintained short-term hydration levels and avoided excessive dehydration in artistic preadolescent and adolescent gymnasts. A higher amount of fluid, equivalent to about 1.5 times the fluid loss, did not provide an additional performance benefit.

## 1. Introduction

It is documented that dehydration is associated with decrements in various fundamental cognitive abilities involving short-term memory, working memory, and fine visual–motor abilities, with the magnitude of the performance decrease in proportion to the level of dehydration [1,2]. According to a recent meta-analysis [3], loss of body fluids at ≥2% of body mass has been associated with significant impairments, particularly for tasks involving attention, concentration, reaction time, and motor coordination, as well as for extremely critical tasks, and especially for sports that require a high level of physical ability and technical skill and utilize complex and intense short movements, such as gymnastics. Gymnasts demand numerous athletic characteristics, which, among others, include strength, power, agility, flexibility, coordination, and balance. Particularly, high levels of executive attention are required to perform the optimal exercise routine during training sessions and official competitions [4]; thus, factors that might negatively affect the aforementioned characteristics, possibly including body water deficit, should be cautiously monitored.

Moreover, although recent data clearly indicate that hypohydration at >2% of body mass loss impairs performance, cognition, and technical skills in both team and individual sports [5,6], especially under heat stress, there are limited data concerning adolescents, and especially for a hardly ever studied athletic population such as male gymnasts. To the best of the authors’ knowledge, there is only one study, conducted by Arnaoutis and his colleagues [7], that investigated the hydration status of elite young gymnasts during a training day. Their main findings were that the prevalence of hypohydration among the elite young athletes of different sports was very high, and that most of the athletes remained dehydrated throughout a typical training day and were dehydrated even more during their practice sessions, notwithstanding ample fluid availability in the field. Considering the fluid deficit observed in the study, the question that arises is to what extend ad libitum fluid intake may or may not produce a considerable fluid loss during training. Furthermore, another question could be whether prescribed fluid intake provides a benefit in terms of fluid balance and performance.

Therefore, the purpose of the present study was twofold: (a) to examine the hydration status of young male gymnasts during a typical training session, and (b) to evaluate, in cases of a fluid deficit during training, the possible influence of a higher amount of prescribed fluid intake on the artistic gymnastic performance.

## 2. Materials and Methods

### 2.1. Experimental Design

Fluid loss in training was estimated during three identical 3 h training sessions, separated by 3–7 days, during which gymnasts drank water ad libitum. After determining fluid loss, gymnasts performed two 3 h sessions that were identical to the ad libitum training sessions separated by one week, while they ingested, in a random order, artificially sweetened water equivalent to either 50% (Low Volume (LV)) or 150% (High Volume (HV)) of the predetermined fluid lost. After the 3 h training at both LV and HV, participants performed the same program routines on three apparatuses and were evaluated by an international-level judge via the assistance of video analysis. Dietary and training statuses were controlled before the LV and HV conditions, which were performed under similar environmental conditions.

### 2.2. Subjects

The study included eleven male competitive-club-level artistic gymnasts: aged 12.3 ± 2.6 years; pubic hair/genital development: 2–3 (mode value; [8]); body mass: 40.64 ± 14.03 kg; height: 144 ± 18 cm; BMI: 19.0 ± 2.4 kg·m^2^; volunteered in the present study. The participants were members of an artistic gymnastics club and trained 6 days per week for about four hours per day. Before the commencement of the study, gymnasts and their parents were thoroughly informed about its purpose, the experimental procedures, any discomfort, and the possible risks involved, and parents signed an inform consent approving their children’s participation. The study was registered in the OSF Registries (https://osf.io/z5xeh (accessed on 30 May 2023)).

### 2.3. Procedures

#### 2.3.1. Ad Libitum Fluid Intake Training Sessions

To determine fluid loss during training, three identical training sessions were performed during which participants ingested water ad libitum. All training sessions started at 18:00 h in a gymnasium under thermoneutral conditions (21–23 °C and 50–65% relative humidity) and consisted of 5 min running for warm-up (at about 120 bpm), 15 min stretching exercises, 20 min general physical conditioning exercises, followed by a series of artistic gymnastics routines on all 6 male apparatuses in a circuit manner, interspersed by specific strength and flexibility exercises.

Before the start of each session, body weights of the young athletes were measured and recorded on a sensitive digital scale accurate to ±0.02 kg (PS 400LBAT; Delmac Instruments, Athens, Greece). During the training sessions, gymnasts’ rates of perceived exertion (RPEs) [9], abdominal discomfort (AD), gut fullness (GF), and thirst (Th) were also evaluated using appropriate scales. Regarding the AD, GF, and Th scales, these ranged from 0 (AD: “Completely comfortable”; GF: “Empty”; Th: “Not thirsty at all”) up to 10 (AD: “Unbearable pain”; GF: “Bloated”; Th: “Extremely thirsty”).

After completing the 3 h training session, each participant performed a program on 3 different apparatuses—floor exercise, parallel bars, and high bar—that was the same in all 3 sessions for all volunteers. The start values of the routines are presented in Table 1. Τhe final score received by each athlete was the sum of the start value plus the execution score, which had a maximum value of ten points in the case of a perfect execution of the exercises.

Afterwards, body weights of the gymnasts were recorded as described above.

#### 2.3.2. Prescribed Fluid Intake Training Sessions

One–two weeks after completing the last ad libitum fluid intake training session, volunteers performed the LV and HV conditions. These training sessions were also identical to the ones previously performed in which fluid was drunk ad libitum. Both LV and HV experimental conditions were conducted on the same weekday with 7 days apart. Participants arrived at the gymnasium at 17:30, emptied their bladders, and baseline urine samples were collected. After this, each gymnast’s nude body mass was measured as described above, and each participant dressed into his sporting gear, which was the same in both the LV and HV conditions. Then, volunteers wore heart rate monitors (Polar FS2c, Kempele, Finland) for recording heart rate (HR) responses during exercise. The HR responses were recorded every 15 min, whereas the RPE, AD, GF, and Th were recorded every 30 min. Exercise started at 18:00, and participants ingested, in a random order, artificially sweetened water that corresponded to 50% (LV) and 150% (HV) of the average fluid loss measured in the three previously performed training sessions. Athletes were instructed to ingest all fluid provided by the end of the 3 h training sessions before performances on the 3 apparatuses took place. Volunteers were told that the aim of the study was to examine the effect of different volume and carbohydrate concentration solutions on performance. Therefore, the artificially sweetened water consisted of tap water and 2 g/L and 4 g/L of a sweetener (Sweet’NLow, Cumberland Packing Corporation, Brooklyn, NY, USA) in the HV and LV conditions, respectively. Urine output was also recorded throughout experimental conditions using an appropriate scale (Philips Essence HR 2394; Philips Budapest, Hungary) and assuming that 1 mL = 1 g. The urine output was also used to estimate sweat loss according to the following formula: Sweat loss = [(BMbefore exercise − BMafter exercise) − urine output] + fluid intake. After the 3 h training, participants performed the same program routines on 3 different apparatuses that they had performed in the ad libitum fluid intake sessions (Table 1). The programs were videotaped (Panasonic NV-GS75, Matsushita Electric Co., Ltd., Osaka, Japan), and performances were evaluated by an international judge who was unfamiliar with the experimental treatments and the purpose of the study.

Gymnasts were evaluated in three of the total six events. The reason was that these three events represent–cover the “different execution positions” on the apparatuses. More specifically, floor exercises are considered a lower limb competition, while parallel bars and the high bar involve the participation of the upper limbs from support and hang positions, respectively. Gymnasts performed various elements according to the rules of the code of points [10] that fulfilled the special requirements of each apparatus. A global description of the routines is shown in Table 1.

As soon as a gymnast completed the performance section, a urine sample was collected, and nude body weight was also recorded. Urine samples were stored at 6–8 °C and analyzed within one hour after collection of the last sample for specific gravity (USG), using a digital urine specific gravity refractometer (ATAGO UG-α, Saitama, Japan). Relative humidity and ambient temperature were monitored every hour (Brannan, Cumbria, UK) and were found to be similar under LV and HV conditions (average temperature: 23.4 ± 1.1 °C in both LV and HV; average relative humidity: 54 ± 4% in LV and 53 ± 5% in HV).

#### 2.3.3. Diet and Training Control

Participants weighed (Kenwood chef/major kitchen scale, United Kingdom) and recorded their normal food intake the day before the first condition (LV or HV) and the day of the condition up to the initiation of exercise, and they were asked to replicate this diet for the same period before the second condition. Volunteers were also requested to record their water intake on the day of the LV or HV up to the initiation of exercise. Furthermore, athletes ingested the last meal or snack on the day of the LV and HV two hours before exercise and were advised to drink 6 mL·kg·BM^−1^ water to facilitate euhydration [11]. Dietary records were analyzed based on published data [12] and food labels. Moreover, gymnasts performed similar training 2 days before the LV and HV and did not train the day before the LV and HV.

#### 2.3.4. Statistical Analyses

Data were analyzed using SPSS (SPSS, Inc., Chicago, IL, USA v. 22.0). The Shapiro–Wilk test was used to assess normality of data. A repeated-measures two-way ([treatment: LV vs. HV] × time [time points]) analysis of variance (ANOVA) was used to compare BM, HR, AD, GF, Th, RPE, and USG responses. Dietary intake, water intake on the day of main conditions, and average GF were analyzed using the Wilcoxon Signed Rank test, whereas fluid loss, estimated sweat loss, urine volume, changes in BM, average HR, AD, Th, RPE during the 3 h exercise period, as well as average performance on the 3 apparatuses were analyzed using a two-tailed paired *t*-test. Performance data were analyzed not only by treatment (LV vs. HV), but also by order (Condition 1 vs. Condition 2). When a significant interaction in the two-way ANOVA was found, simple main effects were used with Bonferroni adjustment for multiple comparisons. All assumptions associated with repeated-measures designs were tested, and the degrees of freedom for significant main effects, interaction, and error term were adjusted according to Greenhouse–Geisser epsilon when the assumption of sphericity was violated. Effect size for main effects and interaction was estimated by calculating partial eta squared (η^2^) in the ANOVA, Cohen’s d (d) in the *t*-tests, and r in the Wilcoxon Signed Rank test. In case of significance, 95% confidence intervals (CI) of the mean (ANOVA and *t*-test) and median (Wilcoxon) differences are also reported. To estimate the sample size, G*Power v. 3.1.9.7 was used. The USG value reported before training in adolescent gymnasts [7] was used (a USG of 1.015, which is considered within the euhydrated range [11]) as a result of fluid intervention, and a standard deviation of 0.01 to account for individual variability [7,13]. Using the above parameters, a large effect size of 1.0, a power of 0.8, and a probability level of 0.05 revealed an estimated sample size of 10. Data are presented as mean ± SD, and the level of significance was set at α < 0.05.

## 3. Results

### 3.1. Ad Libitum Fluid Intake Training Sessions

The average fluid intake ad libitum in the three sessions was 566 ± 176 mL (range: 301–944 mL). Without accounting for fluid intake, BM decreased by 0.44 ± 0.19 kg (range: 0.21–0.88 kg), which corresponded to 1.1 ± 0.3% (range: 0.6–1.5%) of BM. When fluid intake was accounted for, BM changed, decreasing by 1.01 ± 0.29 kg (range: 0.64–1.54 kg), corresponding to 2.6 ± 0.4% (range: 1.9–3.3%) of BM. Thus, the average fluid intake ad libitum corresponded to about 56% of the total fluid lost [(0.566/1.01) × 100 = 56%]. In other words, the gymnasts drank ad libitum about half of the fluid lost during the three identical training sessions.

### 3.2. Body Mass Changes, Fluid Losses, USG, Urine Volume, and Estimated Sweat Loss in LV and HV Conditions

Body mass changes, fluid losses, USG, urine volume, and estimated sweat loss during exercise in both the LV and HV conditions are presented in Table 2. The two-way ANOVA for BM revealed significant interaction (i.e., Fluid × Time) (F_1,10_ = 7.42, *p* = 0.021, η^2^ = 0.43). In the HV condition, BM was maintained after exercise (*p* = 0.36), whereas in the LV condition, BM was lower (*p* < 0.001) post-exercise compared to BM before exercise (Table 2). As a result, the fluid loss was higher in the LV condition compared to the HV condition (*p* = 0.02). Similar results were observed for fluid losses expressed as a percentage of BM (% BM change), or when fluid losses were corrected for fluid intake (Table 2).

Regarding the USG, the two-way ANOVA showed a difference at the interaction level (F_1,10_ = 31.97, *p* < 0.001, η^2^ = 0.76). The pre-exercise USG was similar between conditions. However, the USG was reduced post-exercise in the HV condition (*p* < 0.001), but in the LV condition, the USG after exercise was similar to that pre-exercise, and higher than the corresponding values in the HV condition (Table 1). Furthermore, the urine volume and estimated sweat loss were about 3.4 times and 8% (*p* = 0.03) higher, respectively, in the HV condition compared to the LV condition. Nevertheless, there was considerable variability among the gymnasts in all the above responses, as indicated by the range of values.

### 3.3. Heart Rate, Rate of Perceived Exertion, Abdominal Discomfort, Gut Fullness, Thirst, and Environmental Conditions in LV and HV Conditions

The two-way ANOVA for HR revealed significant differences only at the time level (F_11,110_ = 2.59, *p* = 0.006, η^2^ = 0.21), and the mean HR during exercise was similar between conditions (HV: 122 ± 12 vs. LV: 124 ± 19 b·min^−1^, *p* = 0.49, *d* = 0.22).

Moreover, for the RPE, no difference was found at any level, and the average RPE responses throughout exercise were similar between conditions (HV: 12 ± 3 vs. LV: 12 ± 3, *p* = 0.62, *d* = 0.16). In terms of AD, a difference was observed at the fluid level (F_1,10_ = 5.60, *p* = 0.039, η^2^ = 0.36), and a tendency for significance at the time level (F_2.5,25.4_ = 2.81, *p* = 0.07, η^2^ = 0.22), but the average AD responses were not different (HV: 3.2 ± 1.9 vs. LV: 2.5 ± 1.3, *p* = 0.09, *d* = 0.57). Regarding GF, a difference was found at the time level (F_2.5,25.4_ = 3.72, *p* = 0.03, η^2^ = 0.27), and a tendency for significance at the fluid level (F_1,10_ = 4.30, *p* = 0.07, η^2^ = 0.30), while the average GF was not different between conditions (HV: 1.9 ± 1.6 vs. LV: 1.1 ± 0.9, *p* = 0.07, r = 0.39). In terms of Th, differences were observed at the fluid (F_1,10_ = 7.12, *p* = 0.02, η^2^ = 0.42) and time (F_6,60_ = 2.77, *p* = 0.02, η^2^ = 0.22) levels, whereas the average Th response was higher in the LV condition compared to the HV condition (HV: 3.6 ± 1.4 vs. LV: 5.2 ± 2.1, CI: from −2.8 to −0.3, *p* = 0.02, *d* = 0.85). Finally, both conditions were conducted under almost identical temperatures (HV: 23.4 ± 1.1 vs. LV: 23.4 ± 1.1 °C, *p* = 0.81, r = 0.05) and humidity (HV: 53 ± 5% vs. LV: 54 ± 4%, *p* = 0.45, r = 0.16) conditions.

### 3.4. Dietary Control

The analysis of the dietary data showed that there were no differences between the LV and HV conditions in the daily energy (HV: 2032 ± 502 kcal vs. LV: 1989 ± 597 kcal, *p* = 0.72, r = 0.09) or carbohydrate (HV: 211 ± 68 g vs. LV: 215 ± 70 g, *p* = 0.59, r = 0.13), fat (HV: 92 ± 35 g vs. LV: 85 ± 36 g, *p* = 0.07, r = 0.43), or protein (HV: 102 ± 31 g vs. LV: 95 ± 30 g, *p* = 0.14, r = 0.34) intake consumed the day before each condition. In addition, on the day of, the LV and HV conditions, the energy (HV: 1310 ± 361 kcal vs. LV: 1279 ± 381 kcal, *p* = 0.72, r = 0.09), carbohydrate (HV: 121 ± 40 g vs. LV: 117 ± 42 g, *p* = 0.72, r = 0.09), fat (HV: 63 ± 25 g vs. LV: 61 ± 26 g, *p* = 0.72, r = 0.09), protein (HV: 73 ± 30 g vs. LV: 71 ± 28 g, *p* = 0.46, r = 0.17), and water intake (HV: 622 ± 379 mL vs. LV: 695 ± 443 mL, *p* = 0.29, r = 0.26) from the morning up to the initiation of exercise were similar between conditions.

### 3.5. Performance in LV and HV Conditions

The individual sums of the performance data on the three apparatuses are presented in Figure 1. No difference was observed between the conditions in the sum of the performance (HV: 26.05 ± 2.00 vs. LV: 26.17 ± 2.04; CI: −0.61–0.35, *p* = 0.57, d = 0.18), nor in the average performance (HV: 8.68 ± 0.67 vs. LV: 8.72 ± 0.68; CI: −0.20–0.12, *p* = 0.56, d = 0.18). Moreover, when the performance was analyzed by order (i.e., Condition 1 vs. Condition 2), no difference was found in the sum of the performance (Condition 1: 25.99 ± 2.02 vs. Condition 2: 26.23 ± 2.01; CI: −0.69–0.22, *p* = 0.28, d = 0.35), nor in the average performance (Condition 1: 8.67 ± 0.67 vs. Condition 2: 8.74 ± 0.67; CI: −0.23–0.08, *p* = 0.29, d = 0.34) achieved.

## 4. Discussion

The aims of the current study were, first, to examine the hydration status of young preadolescent and adolescent male gymnasts during a typical training session and evaluate, in cases of fluid deficit during training, the possible influence of a higher amount of prescribed fluid intake on the artistic gymnastic performance. Our main findings were as follows: (i) during a typical 3 h training session in a thermoneutral environment, young artistic preadolescent and adolescent gymnasts drank ad libitum about half of their fluid lost; (ii) the ingestion of about 50% of the fluid loss during training sustained the body weight loss below the recommended threshold of −2% of body mass; (iii) prescribed hydration protocols maintained euhydration during training; and (iv) the consumption of fluids reaching up to 150% of the fluid loss were well tolerated during training but did not provide any additional performance benefits.

The results from the present field research are in accordance with previous published studies [14,15] that indicate that prescribed drinking is an effective method of assuring euhydration status, avoiding parallelly significant decrements in athletic performance. More precisely, de Landa and colleagues [15], who tested the same athletic population as the one in the present research but at a higher age (about 18 years), highlight the amelioration of hydration intake and, consequently, performance in specific technical skill exercises via the use of an individualized hydration pattern compared to a habitual drinking one.

To the best of our knowledge, there is only one study that has also assessed the hydration status of young elite preadolescent and adolescent gymnasts [7]. In this later study, however, the young gymnasts consumed fluids ad libitum and experienced serious levels of dehydration during their training; therefore, programmed drinking seems preferable to the suggestion of “drinking according to thirst” proposed in several studies [16]. Moreover, national organizations, such as the American College of Sports Medicine, and pioneers in the field of sport nutrition highlight the importance of an individualized fluid plan, which encourages fluid intake, especially in high-performance sport, where fluid mismatches appear to be common [11,17]. Considering the aforementioned, prescribed personalized fluid plans should be applied to the athletes, taking into consideration the unique characteristics of each event, environmental conditions, practical features, and previous experiences [17].

Another interesting finding is that the consumption equal to 50% of fluid losses measured during training, which corresponded to about the fluid drunk ad libitum by the gymnasts, managed to avoid dehydration at the established threshold of −2% of BW that hampers athletic performance [11]. Interestingly, the ingestion of fluids at a level of 150% of body water loss did not result in any performance benefit in the program routine of the gymnasts, although it provoked better maintenance of hydration status. A possible explanation might be that gymnasts, due to the nature of the sport and to the short durations that the exercises demand, need smaller amounts of fluid. However, the replenishment of the fluid losses of the gymnasts is clearly important, although the intra-individual differences, as demonstrated by the range of values in Table 2, suggest that a personalized rehydration approach should be implemented, and not just general advice provided. Numerous parameters, such as the personalized sweating rate, exercise intensity and duration, and availability of fluids during training, among others, clearly can affect the results and should be carefully monitored and taken into consideration [18]. In practice, monitoring the fluid losses of each athlete regularly in training and at different periods of the year, during which the temperature may vary, will establish individualized fluid needs and form specific hydration approaches that can be applied on an individual basis.

Despite the considerable amount of fluid provided during the HV condition (1509 mL), which corresponded to about 3.7% of their BM, the young gymnasts experienced no significant gut fullness (about 2 = “almost emptied”) or abdominal discomfort (about 3 = “very comfortable” to “somewhat comfortable”) during training, nor any detrimental effect in terms of performance compared to the volume drunk in the LV condition that was in fact their ad libitum fluid intake as recorded in the ad libitum training sessions. This demonstrates that young artistic gymnasts can drink fluid that considerably exceeds that of their fluid lost during training, maintaining an even better fluid balance without experiencing abdominal discomfort or compromising performance.

Finally, it is obvious that continued efforts must be made by coaches, parents, and athletes to educate young athletes about the benefits of achieving an optimal hydration state, the development of more efficient and realistic hydration strategies, and the methods of accurately assessing and monitoring their hydration status [19,20].

A limitation of the study was the fact that the hydration status was evaluated only shortly after the completion of exercise. It would have been interesting to examine the hydration status of the participants the next morning or, even better, at the start of the next day’s training session. However, this approach requires further control over diet and physical activity. Ideally, a longer intervention period, such as one in which the fluid balance is monitored, would possibly show a long-term effect.

Furthermore, performance might have been evaluated on all six apparatuses. Unfortunately, that would have expanded the time of the investigation and that was not feasible due to the availability of the gym facilities.

## 5. Conclusions

In conclusion, ingesting fluid equivalent to about 50% of the fluid lost during a 3 h training session in a thermoneutral environment, an amount that the participants in fact drank ad libitum, maintained short-term hydration levels in artistic preadolescent and adolescent gymnasts and avoided excessive dehydration (>2%). A higher amount of fluid, equivalent to about 1.5 times the fluid loss, did not cause any abdominal discomfort but provided no additional benefit in terms of performance.

## Figures and Tables

**Figure 1 nutrients-15-02667-f001:**
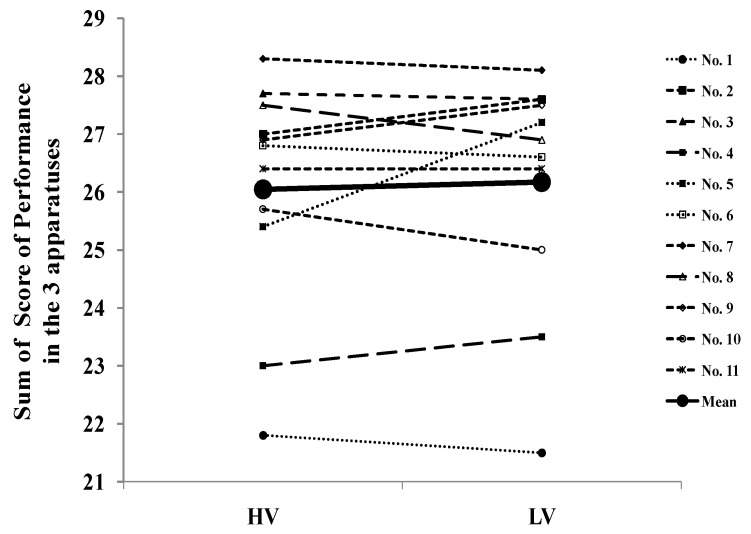
Individual sums of scores of performance data on the three apparatuses in LV and HV conditions. The thick solid line indicates the mean values.

**Table 1 nutrients-15-02667-t001:** Brief description of the exercises on the apparatuses.

Apparatus	Description of Elements	Start Value * (Mean ± SD)
**Floor Exercise**	Saltos forward with (360°, 540°, 720°) or without turns Saltos backward with (540°, 720°, 1080°) or without turns Nonacrobatic elements (Nakayama, Fedorchenko, Scale) Dismount	1.74 ± 0.51
**Parallel Bars**	Elements in support or through support on 2 bars, such as Bilozerchev–Peters, Diamidov, Morisue, Yamavaki, Back uprise. Tippelt, Moy, Kenmotsu, Giant swing, Dismount	1.87 ± 0.67
**High Bar**	Giant swings forward or/and backward with or without turns Flight elements (Cuervo, Tkatchev, Jager) In bar and Adler elements, such as Endo, Stalder with or without turns, Dismount	2.11 ± 0.62

* The “Start value” denotes the difficulty of the routine regardless of how well the exercises are performed.

**Table 2 nutrients-15-02667-t002:** Body mass changes, urine specific gravity, urine volume, and estimated sweat loss in HV and LV conditions. Numbers in parentheses indicate ranges of values (mean ± SD).

Variable	HV	LV	Statistics (HV vs. LV)
**BM_Pre-exercise_ (kg)**	40.67 ± 14.16	40.63 ± 14.01 ^1^	*p* = 0.76 CI: from −0.21 to 0.28
**BM_Post-exercise_ (kg)**	40.57 ± 14.37	40.15 ± 13.84 ^1^	*p* = 0.09 CI: from −0.08 to 0.92
**Fluid Loss (kg)**	0.10 ± 0.33 (from +0.46 to −0.48) *	0.48 ± 0.25 (from −0.16 to −0.84)	*p* = 0.02, *d* = 0.82 CI: from −0.70 to −0.07
**% BM change (%)**	−0.4 ± 0.8 (from +0.8 to −1.5) *	−1.2 ± 05 (from −0.5 to −2.3)	*p* = 0.02, *d* = 0.81 CI: from −1.38 to −0.13
**Fluid Loss Corr. (kg)**	1.61 ± 0.31 (1.23–2.31)	0.98 ± 0.36 (0.48–1.57)	*p <* 0.001, *d* = 2.25 CI: 0.44–0.81
**% BM change Corr. (%)**	−4.3 ± 1.2 (2.5–6.0)	−2.5 ± 0.6 (1.7–3.6)	*p* < 0.001, *d* = 1.64 CI: 1.1–2.5
**USG_Pre-exercise_**	1.015 ± 0.007 ^2^ (1.005–1.025)	1.018 ± 0.007 (1.000–1.025)	*p* = 0.09 CI: from −0.07 to 0.001
**USG_Post-exercise_**	1.002 ± 0.003 ^2^ (1.000–1.010)	1.017 ± 0.006 (1.005–1.025)	η^2^ = 0.76 (*Interaction Effect*) *p* < 0.001 CI: from −0.020 to −0.011
**Urine Volume (mL)**	791 ± 339 (298–1230)	230 ± 108 (45–411)	*p* < 0.001, *d* = 1.83 CI: 255–766
**Sweat Loss (mL)**	816 ± 378 (304–1436)	755 ± 358 (271–1325)	*p* = 0.03, *d* = 0.75 CI: 6–116

BM: body mass; CI: 95% lower and upper confidence interval; *d*: Cohen’s d; Fluid Loss Corr.: fluid loss corrected for fluid intake; % BM change Corr: % of BM change corrected for fluid intake; USG: urine specific gravity; η^2^: partial eta squared; * positive signs indicate hyperhydration. ^1^: *p* < 0.001, CI: 0.32–0.65; ^2^: *p* < 0.001, CI: 0.008–0.018.

## Data Availability

The data presented in this study are available on request from the corresponding author. The data are not publicly available due to the fact that data are from pediatric population.

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
