# Peer review of "Young Artistic Gymnasts Drink Ad Libitum Only Half of Their Fluid Lost during Training, but More Fluid Intake Does Not Influence Performance"

_nutrients, 2023, doi:10.3390/nu15122667_

Round 1
Reviewer 1 Report
The work is very interesting although I have some doubts and suggestions:
1. Table 1 describes apparatus and exercises. I suppose that the third column deals with the experts' evaluations of these exercises, but the score in gymnastics may be unknown to the reader. I suggest that this clarification appear either in the text or in a table footnote.
2. One issue that seems important to me is not to use the same type of beverage in the comparison between self-imposed intakes, and those prescribed in the trial. Perhaps it would be useful to carry out all the sessions with the same beverage composition, since experts recommend the use of isotonic drinks in long duration sessions such as those of the trial.
3. It would be interesting to see if the differences are significant in terms of body composition, because if they are, the subjects may not be eating correctly, since the energy and macronutrient needs may be different and yet the intakes do not differ.
Of particular relevance to their work would be the intake of carbohydrates, responsible for the glycogenic deposits and whose differences are associated with differences in body water content.
4. Although some studies deal with it, it would be important to clarify that most of the ACSM, AIS experts... advise against hydration without specific guidelines in athletes who exercise in such long sessions.
Author Response
Reviewer 1
The work is very interesting although I have some doubts and suggestions:
Reviewer’s Comment:
- Table 1 describes apparatus and exercises. I suppose that the third column deals with the experts' evaluations of these exercises, but the score in gymnastics may be unknown to the reader. I suggest that this clarification appear either in the text or in a table footnote.
Authors’ Response:
The reviewer is absolute right that the score in gymnastics may be unknown to the reader. Authors apologize for the numbers under the column “Start Value” in Table 1 of the previous manuscript that by mistake were reported. The correct Start Value numbers are presented in Table 1 as means ± SD in the revised manuscript. This “Start value” denotes the difficulty of the routine regardless of how well the exercises are performed. This explanation has also been incorporated in the revised manuscript (lines 116-117) as a footnote in Table 1 as requested. Furthermore, how the overall score each athlete receives is further explained in the revised manuscript. Please see lines 110-113.
Reviewer’s Comment:
- One issue that seems important to me is not to use the same type of beverage in the comparison between self-imposed intakes, and those prescribed in the trial. Perhaps it would be useful to carry out all the sessions with the same beverage composition, since experts recommend the use of isotonic drinks in long duration sessions such as those of the trial.
Authors’ Response:
Definitely, we agree that ideally the composition of beverages should have been identical in both conditions. However, we took a different approach for several reasons. Firstly, we indented to have an ecological approach where in practice young artistic gymnasts of this level consume tap/bottled water and not an electrolyte sports drink. On this basis, and in order to “control” the psychology of the participants since the different volume between LV and HV conditions could not be blinded, we inform the gymnasts that “the aim of the study was to examine the effect of different volume and carbohydrate concentration solutions on performance” (please see lines 134-135 of the revised manuscript) and that they actually ingested the same amount of carbohydrates in both conditions, but with different concentrations. Therefore, fluids should have different degree of sweetness. Consequently, in the HV a 2 g/L of an artificially sweeter was used instead of 4 g/L in LV. Taking into account that about ½ of a liter was ingested in LV and about 1.5 lit in HV providing 2 g and 3 g of a sweetener respectively, and that the average of body mass of the participants was about 40 kg, authors considered that these differences in terms of macronutrients and electrolytes would have a negligible effect on the main parameters studied, fluid balance and performance. Therefore, we chose to stress on differences in fluid volume and sweetness regarding the fluid formulation of the beverages used.
Reviewer’s Comment:
- It would be interesting to see if the differences are significant in terms of body composition, because if they are, the subjects may not be eating correctly, since the energy and macronutrient needs may be different and yet the intakes do not differ.
Authors’ Response:
Authors have analyzed dietary data by kg per body mass and also examined relative (%) contribution to energy intake of macronutrients the day before and the training/intervention day in both conditions. All these data are presented in the table below, and as it can been observed no significant difference exists between the two conditions considering body mass of participants.
In terms of diet quality, at least for the previous day of intervention that included all day meals, protein intake requirements in terms of g/kg BM fulfilled requirements, while carbohydrate content is somewhat low compared to guidelines for young athletes (Int J Sport Nutr Exerc Metab. 2014 Oct;24(5):570-84. doi: 10.1123/ijsnem.2014-0031). However, diet quality can not be judged only by one-day recording, since longer dietary records up to a 7-day record improve the reliability of the estimate of energy and nutrient intake by athletes (Int J Sport Nutr Exerc Metab. 2003 Jun;13(2):152-65. doi: 10.1123/ijsnem.13.2.152). Furthermore, the purpose of our study had no intention to examine dietary habits of young artistic gymnasts, and diet was recorded to mainly control this factor that could influence our intervention.
|
|
Day Before |
Training Day |
P* Day Before |
P* Training Day
|
|||||
|
|
HV |
LV |
HV |
LV |
HV Vs. LV |
HV Vs. LV |
|
||
|
CHO (g/kg) |
5.6 ± 2.8 |
5.6 ± 2.6 |
3.2 ± 1.8 |
3.0 ± 1.5 |
1.000 |
0.715 |
|
||
|
CHO (%) |
40 ± 11 |
42 ± 11 |
35 ± 12 |
35 ± 11 |
0.109 |
0.715 |
|
||
|
Fat (g/kg) |
2.2 ± 0.9 |
2.0 ± 0.7 |
1.6 ± 0.7 |
1.6 ± 0.8 |
0.068 |
1.000 |
|
||
|
Fat (%) |
40 ± 10 |
38 ± 10 |
43 ± 10 |
43 ± 10 |
0.109 |
0.715 |
|
||
|
Protein (g/kg) |
2.6 ± 0.9 |
2.4 ± 0.7 |
1.8 ± 0.7 |
1.8 ± 0.8 |
0.197 |
0.655 |
|
||
|
Protein (%) |
20 ± 5 |
20 ± 5 |
22 ± 0.5 |
22 ± 0.6 |
0.068 |
0.465 |
|
||
|
Energy (kcal/kg) |
52 ± 19 |
49 ± 15 |
34 ± 5 |
33 ± 5 |
0.854 |
0.593 |
|
||
*Wilcoxon Signed-rank test
Reviewer’s Comment:
Of particular relevance to their work would be the intake of carbohydrates, responsible for the glycogenic deposits and whose differences are associated with differences in body water content.
Authors’ Response:
Considering the above data of carbohydrate intake in g/kg BM, the fact that no difference was found in fluid intake the day of intervention from the morning up to the initiation of training (please see lines 265-266 of the revised manuscript), and that urine specific gravity was not different before exercise between conditions (please see Table 2 of the revised manuscript), it can be hypothesized that body water content was similar between conditions before exercise and diet did not influence fluid balance before intervention.
Reviewer’s Comment:
- Although some studies deal with it, it would be important to clarify that most of the ACSM, AIS experts... advise against hydration without specific guidelines in athletes who exercise in such long sessions.
Authors’ Response:
We thank the reviewer for the comment. Indeed, this is the direction pointed out by several organizations and researchers in the field of hydration. Therefore, a specific paragraph has been added to the discussion section of the revised manuscript. Please see lines 306-313.

Reviewer 2 Report
The essay presented is interesting as a contribution to science, giving alternative information to the established guidelines and that is why i have some doubts:
- - Line 291: bibliographic reference of the autor is missing (15).
- - Several recognized institucions in the world of sports, such as the American College os Sports Medicine (ACSM), advise against hydratation without a specific guideline in long-term sessions, such as the one proposed in their study. I would like to clarify this issue more about not establishing a specific hydratation guideline for the duration and intensity of the exercise in this sport modality.
- - They commented in the discusión that each intensity or modality of sport requires an ideal fluid in each event. What do you mean by that? This question would be related to the previous one.
- - They use the same type of drink in the comparison between the intake they suggest and those prescribed in the essay they present: it would be better for all sessions to have the same drink comparison, isotonic in this case, according to the recommendations of experts in this field.
If they adequately answer these questions, I would have no problem publishing them.
Author Response
Reviewer 2
The essay presented is interesting as a contribution to science, giving alternative information to the established guidelines and that is why i have some doubts:
Reviewer’s Comment:
- Line 291: bibliographic reference of the autor is missing (15).
Authors’ Response:
The bibliographic reference has been added to the revised manuscript. Please see line 296 of the revised manuscript.
Reviewer’s Comment:
- Several recognized institucions in the world of sports, such as the American College os Sports Medicine (ACSM), advise against hydratation without a specific guideline in long-term sessions, such as the one proposed in their study. I would like to clarify this issue more about not establishing a specific hydratation guideline for the duration and intensity of the exercise in this sport modality.
Authors’ Response:
We thank the reviewer for this comment. Indeed, this is the direction pointed out by several organizations and researchers in the field of hydration. Therefore, a specific paragraph has been added to the discussion section of the manuscript. Please see lines 306-313 of the revised manuscript. Moreover, a sentence has been added in the revised version of the manuscript explaining how individualized fluid needs could be established in practice. Please see revised manuscript lines 328-331.
Reviewer’s Comment:
- They commented in the discusión that each intensity or modality of sport requires an ideal fluid in each event. What do you mean by that? This question would be related to the previous one.
Authors’ Response:
This is a very interesting remark from the reviewer. It is well known that each sport has unique characteristics like intensity, duration, position of the body, complexity etc. Moreover, fluid consumption is affected by numerous parameters, like palatability, GI discomfort, breaks, proximity of fluids in the field of play and others thus, a general rule cannot be applied for every activity. However, the main key-points derived from our present study are that: (i) fluid intake should be individualized, and (ii) that even fluid equivalent to about 50% οf fluid loss was enough to avoid excessive dehydration (>2%). The later seems reasonable, especially in a sport like gymnastics where body position and the difficulties of every exercise have a great impact on the athlete’s gastrointestinal system.
Reviewer’s Comment:
- They use the same type of drink in the comparison between the intake they suggest and those prescribed in the essay they present: it would be better for all sessions to have the same drink comparison, isotonic in this case, according to the recommendations of experts in this field.
Authors’ Response:
Definitely, we agree that ideally the composition of beverages should have been identical in both conditions. However, we took a different approach for several reasons. Firstly, we indented to have an ecological approach where in practice young artistic gymnasts of this level consume tap/bottled water and not an electrolyte sports drink. On this basis, and in order to “control” the psychology of the participants since the different volume between LV and HV conditions could not be blinded, we informed the gymnasts that “the aim of the study was to examine the effect of different volume and carbohydrate concentration solutions on performance” (please see lines 134-135 of the revised manuscript) and that they actually ingested the same amount of carbohydrates in both conditions, but with different concentrations. Therefore, fluids should have different degree of sweetness. Consequently, in the HV a 2 g/L of an artificially sweeter was used instead of 4 g/L in LV. Taking into account that about ½ of a liter was ingested in LV and about 1.5 lit in HV providing 2 g and 3 g of a sweetener respectively, and that the average of body mass of the participants was about 40 kg, authors considered that these differences in terms of macronutrients and electrolytes would have a negligible effect on the main parameters studied, fluid balance and performance. Therefore, we chose to stress on differences in fluid volume and sweetness regarding the fluid formulation of the beverages used.
If they adequately answer these questions, I would have no problem publishing them.
